# Population-scale study of eRNA transcription reveals bipartite functional enhancer architecture

Katla Kristjánsdóttir [1,3], Alexis Dziubek [1,3], Hyun Min Kang [2✉] & Hojoong Kwak [1,2✉]

Enhancer RNAs (eRNA) are unstable non-coding RNAs, transcribed bidirectionally from active regulatory sequences, whose expression levels correlate with enhancer activity. We use capped-nascent-RNA sequencing to efficiently capture bidirectional transcription initiation across several human lymphoblastoid cell lines (Yoruba population) and detect ~75,000 eRNA transcription sites with high sensitivity and specificity. The use of nascent-RNA sequencing sidesteps the confounding effect of eRNA instability. We identify quantitative trait loci (QTLs) associated with the level and directionality of eRNA expression. High-resolution analyses of these two types of QTLs reveal distinct positions of enrichment at the central transcription factor (TF) binding regions and at the flanking eRNA initiation regions, both of which are associated with mRNA expression QTLs. These two regions—the central TF-binding footprint and the eRNA initiation cores—define a bipartite architecture of enhancers, inform enhancer function, and can be used as an indicator of the significance of non-coding regulatory variants.

[1] Department of Molecular Biology and Genetics, Cornell University, Ithaca, NY 14853, USA. [2] Department of Biostatistics, University of Michigan, Ann Arbor, MI 48109, USA. [3] These authors contributed equally: Katla Kristjánsdóttir, Alexis Dziubek. ✉email: hmkang@umich.edu; hk572@cornell.edu

Regulation of transcription is achieved mainly through the binding of transcription factors (TFs) at transcription regulatory elements (TREs), such as promoters and enhancers. Genes are expressed from promoters, which integrate regulatory signals from proximal and distal enhancers to determine the amount of RNA product. Such regulatory networks are key to most cellular processes, including development, cell-type differentiation, and stress response. Their misregulation can often cause diseases, and disease-associated genetic variation is enriched in TREs[1–3]. Therefore, considerable efforts have been made to connect genetic variations to molecular phenotypes at TREs, and to understand how those might affect gene expression[4–11].

Enhancer transcription arises in addition to the target promoter activation in diverse species, including flies and humans[12–16], and its levels track with enhancer activity[16,17]. A pair of enhancer RNAs (eRNAs) are typically transcribed in opposite directions from core transcription-initiation regions flanking the central transcription-factor-binding site (TFBS) of the enhancer[17,18]. While the production of eRNAs has been used to identify active enhancers across numerous cell types and tissues[15], the precise roles of eRNAs in gene regulation have not yet been elucidated[19]. Complicating eRNA detection and quantification is the fact that they are rapidly degraded[15]. This issue makes the use of methods that rely on steady-state RNA, such as RNA-seq or Cap Analysis of Gene Expression (CAGE), less suitable. Nascent-RNA-sequencing methods, such as Precision nuclear Run-On sequencing with 5′-capped (m7G) RNA enrichment[17,20,21] (PRO-cap), overcome this challenge by capturing capped RNA at the synthesis stage.

The mapping of genetic variation to molecular phenotypes at different stages of gene expression has provided important insights into the DNA sequences underlying gene regulation[5–8,22–25]. Similarly, identifying associations between genotypes and enhancer transcription will help connect transcriptional changes at enhancers to changes in gene expression, revealing potential mechanisms for gene regulation. Recent studies have mapped genetic variation to transcription at promoters and enhancers using CAGE, revealing quantitative trait loci (QTLs) associated with alternative promoter usage, promoter shape, and expression[10,11]. However, the statistical power to comprehensively profile enhancer-associated QTLs using CAGE was relatively limited compared to promoter-associated QTLs due to the rapid degradation of eRNAs. Given the properties of PRO-cap that allow it to detect and quantify the transcription of unstable RNAs, we anticipate that a much more comprehensive list of enhancer-associated QTLs can be identified.

This study leverages the variation in transcription initiation at enhancers, measured by PRO-cap, in lymphoblastoid cell lines (LCLs) from 67 individuals. We find thousands of genetic variants that affect either transcription initiation levels (tiQTLs) or the directionality of initiation (diQTLs) at enhancers. We find that these two types of QTLs are enriched at distinct positions within the enhancer architecture. Importantly, both variant types show significant association with mRNA expression, illustrating their potential functionality in gene regulation. Overall, our genetic analyses, investigating the pattern of enhancer transcription, suggest a bipartite architecture of enhancers.

## Results

**PRO-cap reveals transcribed regulatory elements**. We prepared PRO-cap libraries from 69 Yoruba lymphoblastoid cell lines (LCLs) (Supplementary Data 1) for which a large number of transcriptome and chromatin datasets are available[5–8,22,24,26]. We combined all PRO-cap datasets and identified transcribed transcriptional regulatory elements (tTREs), including both enhancers and promoters, with bidirectional divergent nascent transcription within 300 base pairs (bp) of one another (Fig. 1a, Supplementary Fig. 1a). We identified 87,826 tTREs (Supplementary Data 2) with high sensitivity and resolution, as illustrated by examples at two loci (Fig. 1b, Supplementary Fig. 1b). We determined how well our approach identifies Refseq-annotated promoters and EP300 bound enhancers and found that it performs substantially better at identifying EP300-bound enhancers than either nuclear or cytoplasmic CAGE[10,27] (Fig. 1c–g). We also compared our approach to LCL data from another nascent-sequencing method, NET-CAGE[28], and found that PRO-cap identifies EP300-bound enhancers marginally better than NET-CAGE (Fig. 1c–g).

We separated the tTREs into promoters and candidate enhancers based on their transcript stability[17] (CAGE vs PRO-cap; See methods) and their proximity to annotated (Refseq) gene transcription start sites (TSS). Based on the CAGE data, 12,878 tTREs were identified as promoters, and the remaining 74,948 tTREs were identified as enhancers; similar numbers were obtained using Refseq TSSs (Supplementary Data 2). Promoters and enhancers show expected patterns of transcription initiation (Fig. 2a, b), RNA polymerase II (Pol II), H3K27 acetylation, and H3K4 methylation (Fig. 2c, d, Supplementary Fig. 2a–e).

Globally, a genomic annotation test[29] shows tTREs are enriched within accessible DNA regions and contain ENCODE annotated transcription factor binding sites (TFBSs; Fig. 2e). The TFBSs most enriched within tTREs, computed using RTFBSdb30, tend to be cell-type relevant[30] (Supplementary Fig. 2f, Supplementary Tables 1 and 2). In addition, the transcription factors (TFs) associated with the enriched TFBSs are more likely to show evidence of cell-type specificity—exhibiting significantly higher expression in LCLs vs. 53 other cell types - than a control group of all CISBP-motif-associated TFs (Supplementary Fig. 2g).

The tTREs show covariation with nearby mRNA TSSs, indicating their association with nearby genes (Supplementary Fig. 2h). The covariation decreases with distance, decreasing more rapidly downstream of the mRNA TSS than upstream. This is consistent with reports that intragenic enhancers are less activating and can even attenuate expression of their host gene[31].

The identified tTREs are enriched with regulatory genetic variants such as expression quantitative trait loci[22] (eQTLs), chromatin accessibility QTLs[5] (DNaseI-sensitivity QTLs, dsQTLs), and disease associated variants (GWAS SNPs)[32] (Fig. 2f, g, Supplementary Fig. 2i). Interestingly, regulatory variants are more enriched in enhancers than promoters. This result was reproduced using DNase hypersensitive sites (DHSs) to identify enhancers, illustrating that it is not a function of our enhancer identification approach (Supplementary Fig. 2j, k). Together, these results demonstrate the regulatory potential of the identified enhancers, and the superior power of using nascent RNA sequencing to identify them.

**Transcription initiation QTLs enriched at enhancer center**. To investigate how underlying sequences establish the transcriptional signature at tTREs, we tested the association between genetic variation across individuals and the pattern of transcription at tTREs. To avoid confounding effects caused by differences in the read mappability of different alleles, we devised a variant sensitive alignment method that masks out allele-mappability biased regions (see online methods and Supplementary Fig. 3a, b). After allele-mappability masking, we identified 76,630 tTREs (Supplementary Data 2), about 40% of which are variably expressed between individuals (Supplementary Fig. 3c, d).

We identified genetic variants associated with a change in the overall levels of transcription at nearby tTREs (Fig. 3a). We

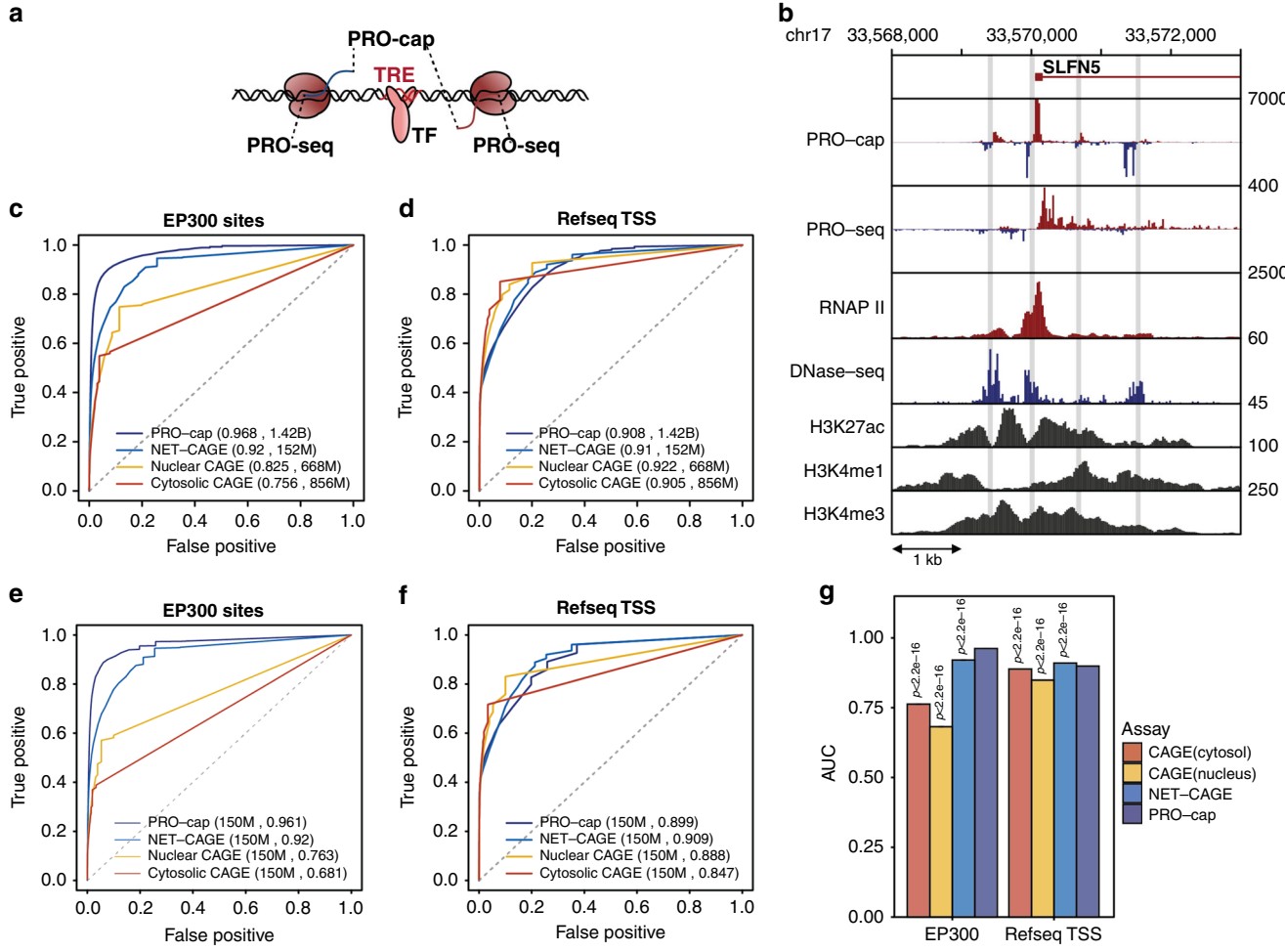

**Fig. 1 PRO-cap identifies tTREs with high resolution and sensitivity. a** Schematic of bidirectional transcription at tTREs. PRO-cap measures nascent-capped-RNA levels and identifies the precise TSS positions (5′ end); PRO-seq measures the 3′ end of RNAs associated with transcriptionally-engaged polymerase. **b** Transcription and chromatin marks at the *SLFN5* locus. PRO-cap, PRO-seq, and DNase-seq data are derived from the YRI LCLs, RNAP II, H3K27, H3K4me2 and H3K4me3 ChIP-seq data are from ENCODE's LCL, GM12878. Shaded regions indicate PRO-cap-identified tTREs. **c** Receiver operating characteristic (ROC) plots of PRO-cap, CAGE from nuclear and cytoplasmic poly-A selected RNA, and NET-CAGE at EP300 bound enhancers (*n* = 18,956). Each profile comes from an aggregate of all replicates. NET-CAGE data is for GM12878 and CAGE data is from ENCODE Riken GM12878. **d** As in **c** for annotated gene promoters (*n* = 12,272). **e** As in **c**, except using downsampling of aggregate datasets to match read counts to 150 million reads per method (*n* = 20 for each dataset). **f** As in **e** for annotated gene promoters. **g** Mean area under ROC curves (**e** and **f**, *n* = 20 downsamplings for each dataset). *P*-values for all tests compared to PRO-cap are by two-sided Welch t-test for the mean AUC for EP300 sites and Refseq promoters. Source data are provided as a Source Data file.

mapped genotypes to quantile-normalized PRO-cap read counts at tTREs within 2 kb and named variants associated with changes in overall PRO-cap signal transcription initiation QTLs (tiQTLs). Overall, we identify 66,906 tiQTLs, and 16,193 tTREs have an associated tiQTL (FDR < 0.1, Supplementary Fig. 4a). We find that 19.2% of tiQTLs are associated with changes in transcription at promoters, which is comparable to the proportion of promoters among tTREs and the proportion of dsQTLs[5] that associate with promoters in the same population.

We examined several parameters to validate our tiQTLs. To verify our results were not due to read-mappability bias, we compared our results with those from a more computationally expensive approach to deal with read-mappability bias (WASP[33]). Around 90% of reads overlapping tiQTLs were correctly remapped to the genome and 87% of tiQTLs on chromosome 22 were reidentified by WASP, verifying that most of our tiQTLs are not the result of allele-biased mapping in our method (Supplementary Fig. 4b–d). Moreover, for tiQTLs located within tTREs, reference and non-reference alleles are equally represented

as the more highly expressed allele (48.7% vs 51.3%), suggesting that reference bias did not play a significant role in tiQTL identification. We also validated our tiQTL results using allele specific expression analysis and estimated the average effect on tTRE transcription initiation to be around 2-fold for the most likely causal SNPs (Supplementary Fig. 4e, f). Overall, the number and quality of QTLs allows us to observe patterns in their location within enhancers and the types of sequences they create or disrupt.

We next explored the link between tiQTLs and the transcriptional architecture of enhancers and promoters. To enrich for primary SNPs, we filtered tiQTLs and kept those with the lowest local minimum *p*-values (within 5 kb), resulting in a reduction to 22,949 tiQTLs. We then examined the tiQTL density around tTRE midpoints (Fig. 3b). At enhancers, the peak of tiQTL enrichment is at the midpoint (Supplementary Fig. 5a), where we would expect transcription factors to frequently bind. Based on these results, we hypothesize that the overall transcriptional activity of enhancers is generally regulated from the central TF

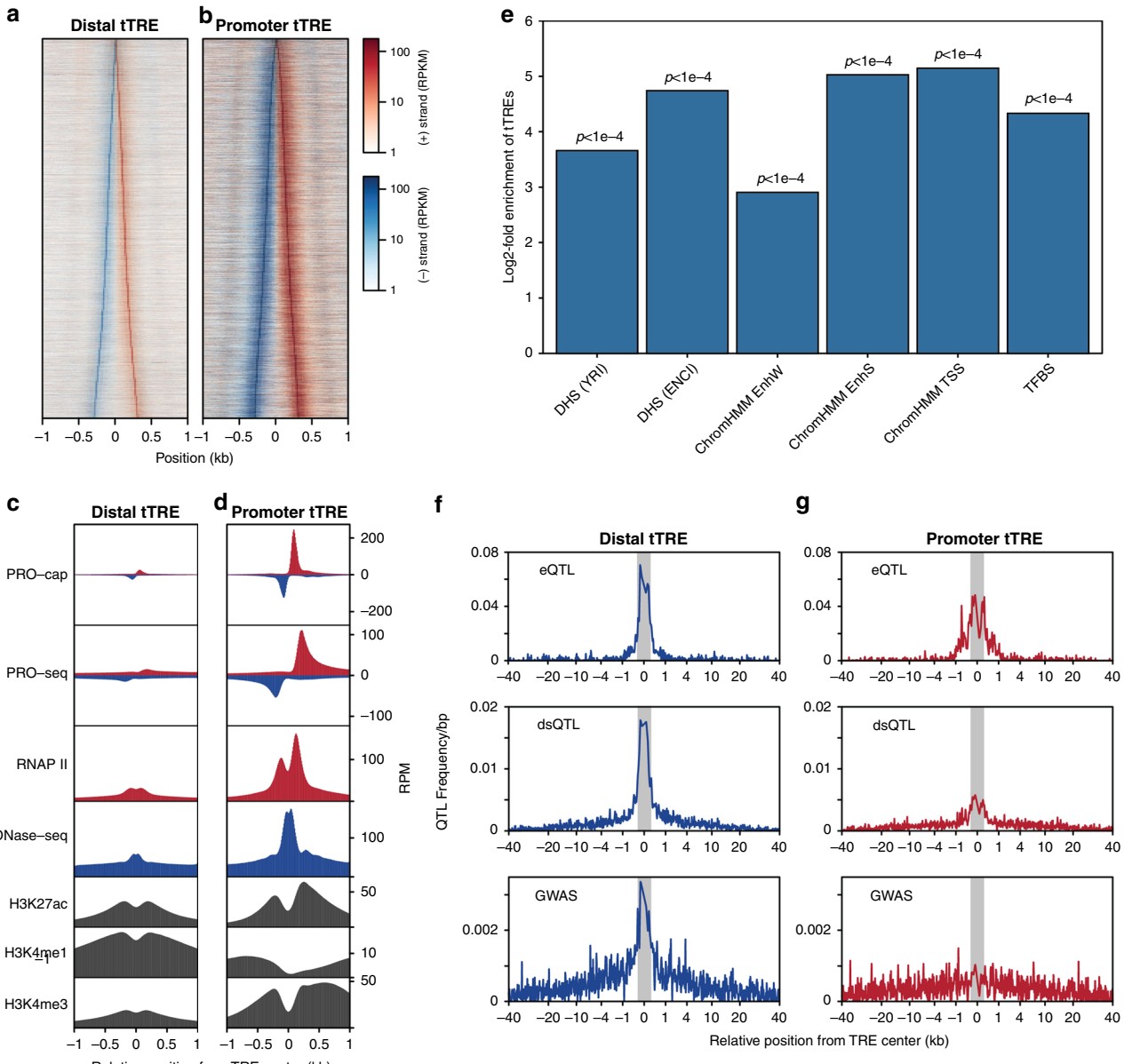

**Fig. 2 tTREs contain important regulatory information. a** PRO-cap signal at enhancers. Heatmap shows plus-strand (red) and minus-strand (blue) read counts at distal tTREs (putative enhancers), ordered by increasing width. **b** As in **a** at promoter tTREs. **c** PRO-cap-identified tTREs have characteristic promoter and enhancer chromatin patterns. Metaplots of PRO-cap, PRO-seq, DNase-seq and RNAP II, H3K27ac, H3K4me1, and H3K4me3 ChIP-seq signals at enhancers (Distal tTRE). **d** As in **c** at promoter tTREs. Promoters are oriented in the direction of the gene. **e** Enrichment of tTREs in regulatory regions. DHS (YRI): DNase I hypersensitive windows from the Yoruba LCLs ($n = 630,168$), DHS(ENC): DNase I hypersensitive sites from ENCODE LCLs (including GM12878; $n = 359,361$), TFBS: defined by ENCODE Factorbook in GM12878 ($n = 212,144$). ChromHMM: regions defined by ChromHMM as weak enhancers (EnhW; $n = 70,620$), strong enhancers (EnhS; $n = 19,362$), and transcription start sites (TSS; $n = 21,342$) in GM12878. GAT computes an empirical, one-sided $p$-value defined as the number of sampled segments that show an equal or greater overlap than the observed overlap. $P$-values are adjusted by Benjamini-Hochberg correction, sampling $n = 10,000$. **f** Frequency of genetic variants associated with gene expression (eQTLs), chromatin accessibility (dsQTLs), and human disease (GWAS) at enhancer tTREs. Both the eQTLs and dsQTLs are derived from the same set of LCLs used in this paper. Shaded regions indicate tTRE boundaries. **g** As in **f** at promoter tTREs.

binding sites. At promoters, we found that tiQTLs are preferentially enriched nearer to the TSSs compared to the tTRE center (Fig. 3c, Supplementary Fig. 5b). Particularly, promoter-associated tiQTLs were enriched at the dominant strand (generally TSS of gene). This difference in tiQTL enrichment between enhancers and promoters agrees with the dual-hub model of promoters[34], and further suggests that the two tTRE types have different tolerances to genetic variation affecting divergent transcription.

To investigate our hypothesis that tiQTLs at enhancers affect TFBSs, we surveyed the underlying sequences. For example, the non-reference allele of the tiQTL *rs185220*, located within a proximal enhancer near the *SETD9* promoter (Fig. 3d), creates a perfect match to the binding site for SP1 transcription factor at the center of the associated enhancer. This alternate allele is also associated with increased eRNA transcription, which is concordant with the change in the TF binding sequence in the central region. We tested the generality of the association

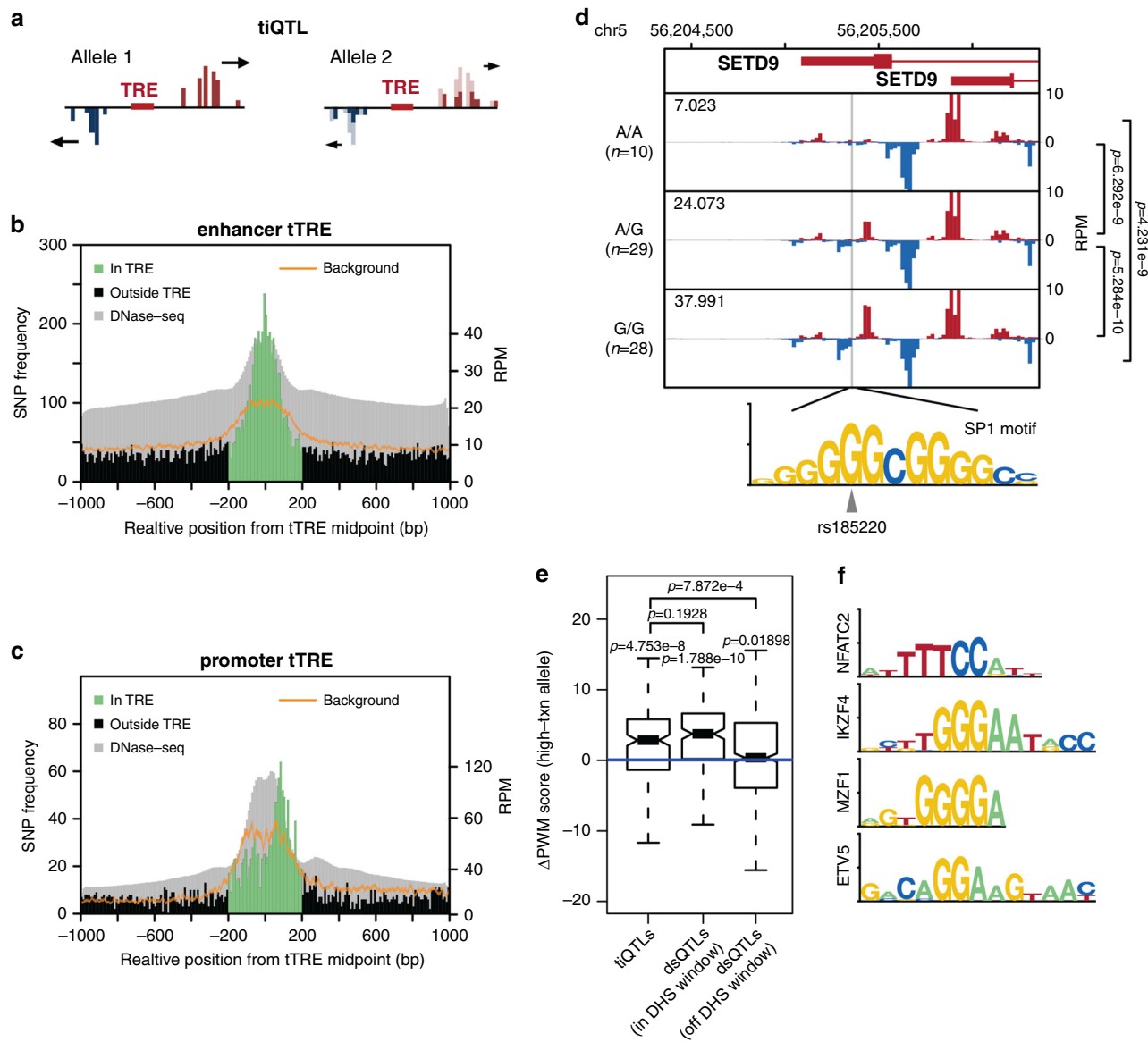

**Fig. 3 Transcription initiation QTLs are enriched in tTREs and affect TFBS motifs. a** Transcription initiation QTL (tiQTL) schematic. **b** tiQTLs are enriched at enhancer midpoints. A histogram of QTL frequency around enhancer midpoints with the expected background distribution with 99% confidence interval (sampled from all SNPs in the same region) shown in orange and aggregate DNase-seq track shown in gray. **c** tiQTLs are enriched at promoter TSSs. As in **b**, at promoters except oriented so that the strand with the dominantly transcribed TSS (usually gene) is downstream of tTRE center. **d** Average PRO-cap signal separated by genotype at tiQTL rs185220. The alternate allele creates a perfect match to the SP1 binding motif. Mean normalized transcription initiation levels for each genotype indicated in the upper left corner. Comparison of transcription initiation index between groups using two-sided Wilcoxon test. Exact $p$-values are $p = 6.3 \times 10^{9}$ between A/A and A/G, $p = 5.3 \times 10^{-10}$ between A/G and G/G, and $p = 4.2 \times 10^{-9}$ between A/A and G/G. **e** Difference in PWM scores between tiQTL high transcription and low transcription alleles with effect size not equal to zero. Center line of boxplot indicates the median, box limits are the 25th and 75th quantiles, whiskers are the 1.5x interquartile range, and the notch reflects the 95% confidence interval of the median. DNase hypersensitivity QTLs are shown for comparison. $P$-values above each group are single sample two-sided Wilcoxon tests. Exact $p$-values are $p = 4.8 \times 10^{-8}$, $p = 1.8 \times 10^{-10}$, and $p = 0.019$. $P$-values between groups are two-sided Wilcoxon tests. Exact $p$ values are $p = 7.9 \times 10^{-4}$ between tiQTLs and off window dsQTLs and $p = 0.19$ between tiQTLs and in window dsQTLs. ($n = 215$, 815, and 603 QTLs respectively). **f** Motifs most often affected by in-tTRE tiQTLs (>20% increase over background).

between tiQTLs and central TF-binding motifs using TFBS position weight matrix (PWM) scores, as described previously[5,35]. On average, the alleles with stronger eRNA transcription have stronger PWM scores than the weaker alleles (Fig. 3e). The effect size is similar to what was observed in QTLs affecting chromatin accessibility (dsQTL), indicating that alteration of TF binding motifs affects both the open chromatin and eRNA transcription. We also identified the motifs most enriched among those affected by tiQTLs (stronger match in

high expression allele) and found that the most enriched motifs are associated with immune-related TFs (Fig. 3f).

**Directionality-associated QTLs enriched at initiation sites.** We identified variants associated with changes in the directionality of divergent bidirectional transcription at tTREs (log₂ ratio of plus-strand reads over minus-strand reads, i.e., directionality index; Fig. 4a) using the same mapping parameters as for tiQTLs. We

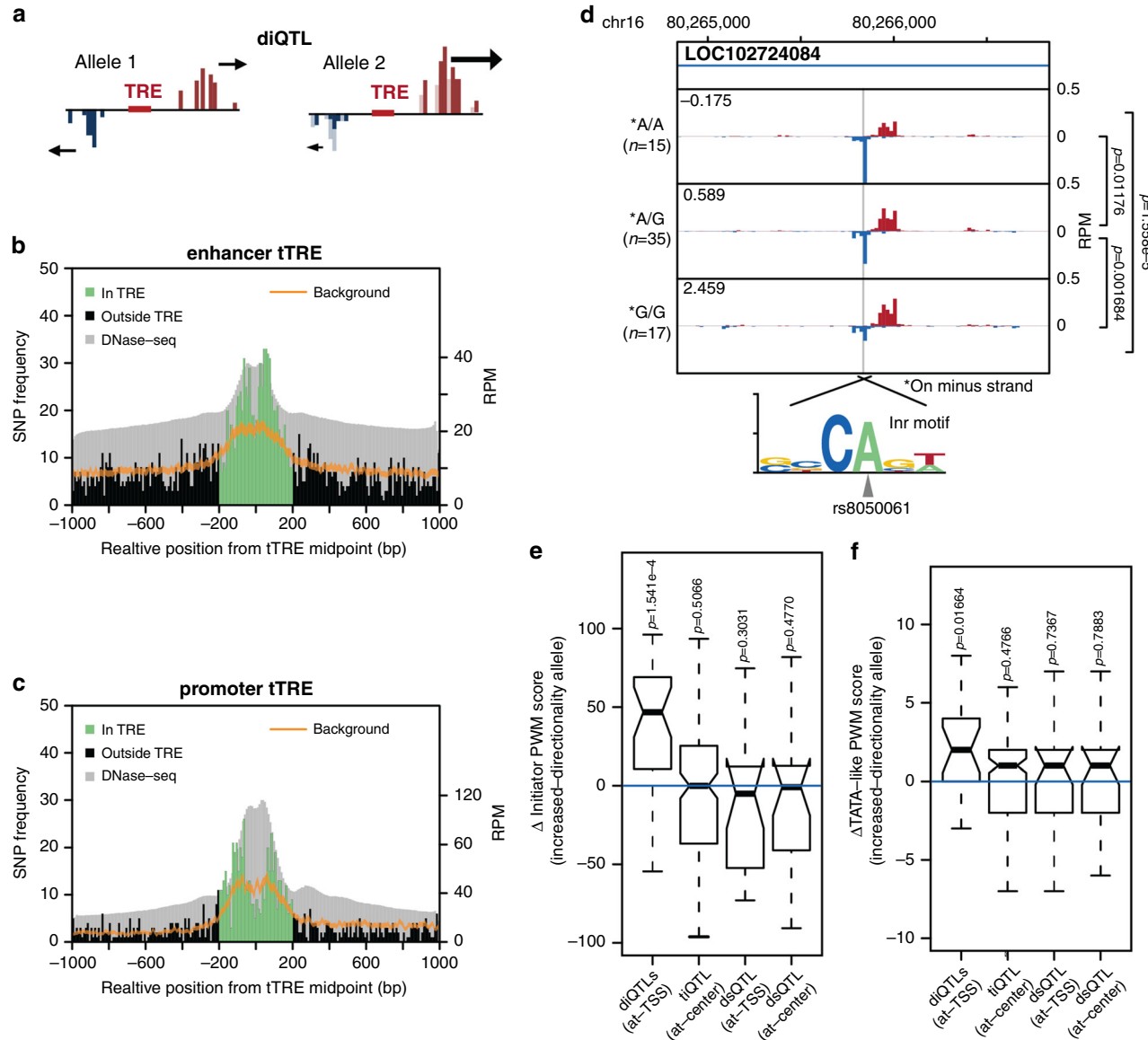

**Fig. 4 Directional initiation QTLs are enriched at TSSs and affect core promoter elements. a** Directional initiation QTL (diQTL) schematic. **b** diQTLs are enriched at enhancer TSSs. A histogram of QTL frequency around enhancer midpoints with the expected background distribution with 99% confidence interval (sampled from all SNPs in the same region) shown in orange and aggregate DNase-seq track shown in gray. **c** diQTLs are enriched at promoter TSSs. As in **b**, at promoters except oriented so that the strand with dominantly transcribed TSS is downstream of the TRE center. **d** Average PRO-cap signal separated by genotype at diQTL rs8050061. The alternate allele disrupts a canonical human Inr motif. Mean directionality index of each genotype indicated in the upper left corner. Comparison of directionality index between groups using two-sided Wilcoxon text. Exact p-values are $p = 0.012$ between A/A and A/G, $p = 0.0017$ between A/G and G/G, and $p = 1.6 \times 10^{-5}$ between A/A and A/G. **e** Difference in Inr PWM score with effect size not equal to zero between increased and decreased directionality allele for diQTLs within 10 bp from the TSSs (at-TSS), with tiQTLs within 10 bp from the tTRE center (at-center) and dsQTLs within 10 bp from either TSS or center included for comparison. Center line of the boxplot indicates the median, box limits are the 25th and 75th quantiles, whiskers are the 1.5 × interquartile range, and the notch reflects the 95% confidence interval for the median. Exact p-values are $1.5 \times 10^{-4}$, $p = 0.51$, $p = 0.30$, and $p = 0.48$ respectively. ($n = 19$, 192, 28, and 27 QTLs respectively). **f** As in **e**, for TATA-box. Here (at-TSS) indicates 40 to 20 bp upstream of TSS, where TATA-box is usually found. Exact p-values are 0.017, $p = 0.48$, $p = 0.74$, and $p = 0.79$ respectively. ($n = 34$, 152, 21, and 22 QTLs respectively).

named these variants directional initiation QTLs (diQTLs), and identified 16,142 of them (Supplementary Fig. 6a). Overall, 4,162 tTREs have an associated diQTL (FDR < 0.1), fewer than the number of tTREs with associated tiQTLs. We found that diQTLs are relatively enriched in promoters; 34.2% of diQTLs are associated with changes in transcription at promoters, while promoters comprise only 16.2% of the total number of tTREs. We also tested the diQTLs for effects of allele mappability bias and saw just under 90% remapping using WASP (Supplementary

Figs. 4b and 6b), indicating that our diQTLs are not artifacts of biased mapping.

SNPs identified as diQTLs and those identified as tiQTLs show significant overlap in terms of the tTREs they are located within. 52.1% of tTREs associated with a diQTL also contain an associated tiQTL, comprising 6.4% of tTREs with tiQTLs. Focusing only on primary diQTL SNPs, defined as described for tiQTLs above, 4,479 primary diQTLs remain within tTREs. Of these filtered diQTL-associated tTREs, 43.6% are also associated

with a tiQTL; this constitutes a 3.76-fold enrichment of tiQTLs compared to a background set of tTREs (background set generated by Broad Institute SNPsnap[36], two-sided Fisher's Exact test $P < 2.2 \times 10^{-16}$). Similarly, tiQTL tTREs are also enriched to have diQTLs (2.65-fold, $P < 2.2 \times 10^{-16}$).

We then examined how diQTLs are linked to the transcriptional architecture of tTREs. Primary diQTL SNPs are most enriched ~70 bp to either side of the enhancer midpoint (Fig. 4b), coinciding with the average relative position of TSSs (Supplementary Fig. 6c). This pattern is distinct from the tiQTL enrichment, which is centered at enhancer midpoints. Based on these results, the directionality of enhancer transcription is mostly regulated in the core-promoter-like regions near the transcription initiation sites, where the pre-initiation complex (PIC) of RNA polymerase II assembles, rather than where transcription factors bind. We found that diQTLs have the same TSS-enrichment pattern at promoters, in this case in agreement with the tiQTL distribution (Fig. 4c, Supplementary Fig. 6d). However, the diQTLs are not preferentially enriched at the dominant strand as seen for tiQTLs. When a variant is identified as both a diQTL and a tiQTL, corresponding to 32.84% of diQTLs and 6.41% of tiQTLs, we see patterns that resemble that for all diQTLs (Supplementary Fig. 6e,f). However, the pattern is less clear, particularly at enhancers.

To explore this difference in the placement of tiQTLs and diQTLs within enhancers, we investigated the underlying sequence of the diQTLs using a similar approach to the one used for tiQTL motifs. We suspected diQTLs would affect sequence elements at regions near the enhancer TSSs. We defined these regions as the core, since they contain core-promoter-like motifs such as the Initiator element (Inr) and the TATA-like elements. For example, the non-reference allele of diQTL rs8050061 disrupts a match to the canonical Initiator element[37] (Inr), coinciding with decreased transcription from the affected strand (Fig. 4d). More globally, Inr is highly enriched at PRO-cap identified TSSs[37] (Supplementary Fig. 6g), and diQTL alleles associated with a shift in directionality towards stronger expression of the strand near which the diQTL is located tend to have stronger matches to the Inr motif (Fig. 4e). To a lesser degree, the same was also true for TATA-like elements, which are usually found 20–40 bp upstream of the TSS and can tolerate a small number of mismatches or positional shifts from the canonical TATA-box motif[38] (Fig. 4f).

**Both ti- and diQTLs correlate with gene expression.** The most critical question when exploring changes in chromatin phenotypes is whether they have an effect on the final output—gene expression. We used RNA-seq data from the same set of LCLs[22] to define expression QTLs (eQTLs) using only SNPs located within 2 kb of a tTRE, to match the set of SNPs used to identify ti- and diQTLs. We found 744 genes with associated eQTLs within 200 kb, with a total of 5,997 eQTLs (FDR < 0.05). Overall, these eQTLs are enriched within 200 bp from the enhancer midpoints defined by PRO-cap (Fig. 5a).

We explored the overlap between tiQTLs, diQTLs, and eQTLs to determine if SNPs associated with enhancer transcription are likely to affect gene expression. Broadly, we found that enhancers that contain an associated tiQTLs and/or diQTL are 4.3-fold and 3.5-fold more likely to also contain an eQTL, respectively, though the overlap is still necessarily low (~3–4%) given the smaller number of eQTLs (Supplementary Fig. 7a). Direct testing of enhancer tiQTLs and diQTLs for association with changes in gene expression reveals a significantly stronger association for both QTL types than for background (Fig. 5b, Supplementary Fig. 7b). Importantly, this is also true for diQTLs regardless of

location within the enhancer tTRE (center or TSS) as well as for diQTLs that are not tiQTLs and therefore do not significantly alter the overall transcription levels (Fig. 5c, Supplementary Fig. 7b).

Further supporting a functional effect, diQTLs and tiQTLs both overlap with other QTLs found in LCLs. Compared to background SNPs, diQTLs, and tiQTLs are enriched for dsQTL and eQTL overlap, as well as several different transcription factor binding QTLs[26] (bQTLs) (Fig. 5d). Enhancer architecture is associated with these overlaps, and the positions of the ti/diQTLs distinguish this. For example, when we filter for diQTLs near the tTRE TSS and tiQTLs near the tTRE center, we see that both QTL classes are enriched for eQTLs and dsQTLs (Supplementary Fig. 7c, d). However, the central tiQTLs are significantly enriched for transcription factor binding QTLs, while the TSS-proximal diQTLs show no enrichment in most of the bQTLs. These results indicate that both tiQTLs and diQTLs are associated with gene expression and, given the different positioning of tiQTLs and diQTLs within enhancers and overlap with bQTLs, that sequences at both the central TFBS and the regions surrounding the TSSs may affect enhancer function in gene regulation through distinct mechanisms.

To further explore the model that both the center and TSS cores are important for enhancer function, we hypothesized that eQTLs would be enriched in those regions relative to the regions outside them. We separated enhancer regions into: the center, the core promoter-like region from which transcription arises (core), the space between them (non-core-non-center, NCNC), the space outside of the core but within the enhancer region (out), and those beyond the enhancer region (far out) (Fig. 5e). Using the far-out region as a baseline, we find a significantly higher eQTL frequency within the center and the core (Fisher's exact test, $P < 0.05$, Fig. 5f). Conversely, the out and NCNC regions have no significant increase in eQTL frequency. Additionally, we compared the eQTL frequency between the core and the out or NCNC regions, normalizing for the distance from the center, using bootstrapping to estimate an empirical significance level (Supplementary Fig. 7e–h). We find that the core regions are significantly enriched with eQTLs compared to the out regions and NCNC regions ($P < 10^{-16}$). These support our hypothesis that the core eRNA initiation regions are indeed important for target gene expression, in addition to the central transcription factor binding regions.

## Discussion

We explored the activity and architecture of transcribed transcriptional regulatory elements (tTREs) by studying the variation in transcription initiation across human LCLs. We identified genetic variants associated with enhancer and promoter transcriptional activity and directionality. The pattern of enrichment for these genetic variants and the types of motifs they affect suggest that overall transcriptional activity at enhancers is regulated from the central transcription-factor binding region (TFBS) and directionality is regulated from the surrounding core initiation regions. Both variant types are associated with gene expression at a higher rate than expected by chance, indicating that both the central TFBS and the flanking core initiation regions affect transcription at enhancers and their role in gene expression. This conclusion is supported by regional enrichment of eQTLs within enhancers.

Identification of enhancers based on capped nascent RNA sequencing provides a direct measure of transcriptional activity and, therefore, shows higher sensitivity than previous methods. A direct measure of transcriptional activity is critical, as non-productive transcripts such as eRNAs are rapidly degraded in the

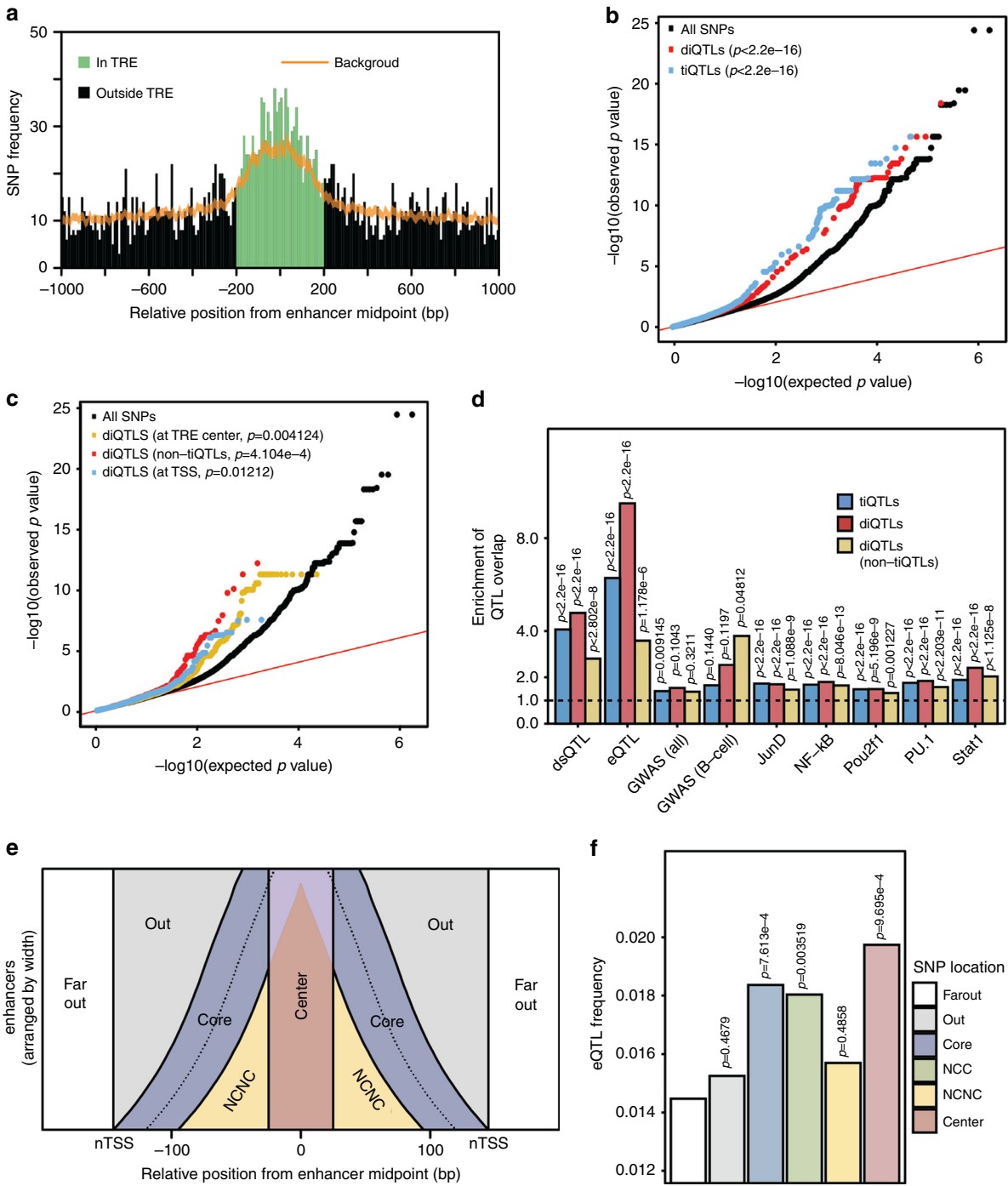

**Fig. 5 ti- and diQTLs at enhancers associate with changes in gene expression. a** eQTLs are enriched at enhancers. A histogram of QTL density around enhancer midpoints with the expected background distribution with 99% confidence interval (sampled from all SNPs in the same region) shown in orange. **b** QQ plot of all *p*-values for each SNP group in eQTL discovery analysis. *P*-values use two-sided Kolmogorov–Smirnov test comparing to the all SNP distribution. **c** As in **b** for subgroups of diQTLs. Exact *p*-values are $p = 0.0041$, $p = 4.1 \times 10^{-4}$, and $p = 0.012$ respectively. **d** Enrichment of ti- and diQTLs in tTREs (±200 bp from center) containing other QTL types. Enrichment is computed compared to all SNPs in the same region. Overlap of QTL and background are compared for each bar with two-sided Fisher's exact test. Exact *p*-values are $p = 2.8 \times 10^{-8}$, $p = 1.2 \times 10^{-6}$, $p = 0.32$, $p = 0.048$, $p = 1.1 \times 10^{-9}$, $p = 8.0 \times 10^{-13}$, $p = 1.2 \times 10^{-3}$, $p = 2.2 \times 10^{-11}$, and $p = 1.1 \times 10^{-8}$ for diQTL (non-tiQTLs) across categories respectively, $p = 5.2 \times 10^{-9}$, 0.10, and 0.12 for diQTLs compared with *POU2F1*, all GWAS, and B-cell related GWAS respectively, $p = 0.0091$ and $p = 0.14$ for tiQTLs compared with all GWAS and B-cell GWAS respectively, and $p < 2.2 \times 10^{-16}$ for all other indicated *p*-values. Source data are provided as a Source Data file. **e** Diagram explaining the separation of candidate enhancer tTREs into functional regions. NCNC: non-center, non-core. **f** eQTL frequency in different enhancer regions. NCNC: non-center-non-core, NCC: non-center-core (core area does not overlap with center area). Each colored bar is compared to Far Out in a two-sided Fisher's exact test. Exact *p*-values are $p = 0.47$, $p = 7.6 \times 10^{-4}$, $p = 0.0035$, $p = 0.49$, and $p = 9.7 \times 10^{-4}$.

nucleus[15]. Other transcription-based approaches, such as CAGE and nuclear short RNA analysis, are impeded by this instability. As we showed, CAGE performs well in identifying promoters but detects enhancers less efficiently than PRO-cap. Additionally, focusing on the bidirectional nascent transcription start sites (nTSSs) using PRO-cap enabled us to filter out spurious transcription from only one strand, which increases the specificity. Although it is possible that enhancer transcription can be unidirectional, our nascent transcription analysis detected mostly bidirectional and divergently paired PRO-cap peaks, and observed very few unpaired peaks.

The fact that genetic variants affecting enhancer directionality are associated with changes in gene expression brings up an interesting question of its mechanism. While the change in directionality towards or away from the sense strand of the gene will obviously affect its expression levels at promoters, how directional initiation at enhancers affects gene expression is less clear. In some cases, modulating the polymerase initiation at only one of the two initiating sites could impact the overall enhancer activity, though in other cases overall transcriptional activity is not affected by the diQTLs. In the latter case, it is possible that, despite the prevailing model that enhancers are orientation-independent, there may be distance and orientation-specific effects on their target genes, potentially involving eRNAs. While our current list of diQTLs is limited due to the sample size, increasing power with a larger scale analysis would allow testing of these possibilities. Additionally, dissecting the putatively directional enhancers identified in this paper, either in reporter assays or through genome editing with CRISPR, and measuring effects on the expression of associated genes could provide important insights into the role of directionality and eRNAs in enhancer function.

Overall, our data show the power of combining capped-nascent-RNA sequencing with human population genetics to explore the architecture of human enhancers. These results suggest a bipartite model for enhancers, where sequences at both the central TFBS and the core promoter regions surrounding the TSSs are important for enhancer function. Our model also supports the hypothesis that eRNA transcription itself can be functional. In practice, this will have important implications for assessing the significance of disease-associated non-coding genetic variants by adding eRNA start sites to the regions of interest.

## Methods

**Preparation of PRO-cap/PRO-seq libraries.** Human lymphoblastoid cell lines (LCLs) were acquired from the Coriell Biorepository. Unrelated individuals were selected from the Yoruba (YRI) population matching those in a study by Degner et al.[5]. (Supplementary Data 1). LCLs were cultured in RPMI 1640 media (Gibco) with 2 mM L-glutamine (Gibco) and 15% Fetal Bovine Serum (Gibco) supplements without antibiotics at 37 °C under 5% $CO_2$. Mid-log phase cultures were seeded at $2 \times 10^5$ cells/ml density in 15 ml suspension culture, and maintained for 24 hr before the cell harvest. Batches of 10 randomly selected cultures were prepared at a time for a total of 100 samples (PRO-cap: 70 individuals + 10 replicates, PRO-seq: 10 individuals + 10 replicates). One individual was dropped out due to unsuccessful PRO-cap library generation and two because they lacked phased genotype data (Supplementary Data 1). Replicate batches, grown from independent cultures, were processed at least 2 months apart. Briefly, LCLs were pelleted by centrifugation at $800 \times g$ for 3 min in 4 °C, washed twice by resuspension in 10 mL 4 °C Phosphate Buffered Saline (PBS) and centrifugation at $800 \times g$ for 3 min, followed by resuspension in 50 μl of storage buffer (50 mM Tris–Cl pH 8.0, 25% glycerol, 5 mM magnesium acetate, 0.1 mM EDTA, 5 mM DTT). Cells were instantly frozen in liquid nitrogen, and stored at −80°C. Upon thawing, cells were incubated in polymerase run-on reactions with ribonucleotide triphosphate (NTP) substrates at a final concentration of 19 μM biotin-11-CTP, 19 μM biotin-11-UTP, 0.19 mM ATP, and 0.19 mM GTP at 37 °C for 3 min. This was followed by nascent-RNA sequencing library preparation[39]. Briefly, biotin-labeled nascent RNA is extracted by Trizol, fragmented through base hydrolysis under 0.1 N sodium hydroxide in 4 °C for 12 min, neutralized by adding Tris–Cl pH 8.0, and passed through P-30 size exclusion column (BioRad). The RNA is bound to streptavidin-magnetic beads

(ThermoFisher), washed 3-5 times, and extracted using Trizol. The purified biotin-labeled RNA is processed through 3′ RNA adaptor (pUGGAAUUCUCGGGUGCC AAGG-/inverted dT/) ligation, 2′nd streptavidin purification, 5′ RNA decapping, 5′ RNA adaptor (GUUCAGAGUUCUACAGUCCGACGAUCNNNNNNNN) ligation, 3′rd streptavidin purification, and reverse transcription (primer sequence: GCCTTGGCACCCGAGAATTCCA). The adaptor sequences are compatible with the Illumina small RNA sequencing platform. cDNA is amplified through 15–20 cycles of PCR amplification using Illumina TRU-seq single barcode multiplexing primers. The PCR product is PAGE purified and subject to high throughput sequencing with the Illumina NextSeq 500.

**Alignment of PRO-cap/PRO-seq reads to the reference genome.** We followed a previously described PRO-seq/PRO-cap data processing procedure[39] and combined alignments for each method (see Supplementary methods for details).

**Identification of nTSSs and tTREs.** We combined all PRO-cap dataset reads and identified ~1.4 billion (1,417,065,796) unique read molecules mapped to the hg19 genome. We then used a bidirectional read-count filtering approach comparable to capped-RNA analysis described elsewhere to identify bidirectional transcribed tTREs[15] (Supplementary Fig. 1a, see Supplementary Methods for details).

**Evaluation of the sensitivity and specificity of PRO-cap.** To assess the predictive power of PRO-cap to find transcriptional regulatory elements, we used the transcription start sites of the genes with greater than 1 RPKM as the TSS standards from the mRNA-seq data in the YRI LCLs[22] (n = 12,272), and ENCODE FACTORBOOK defined EP300 binding sites in GM12878 cell line as the enhancer standards (n = 18,956). We used the randomly selected 1 million genomic regions in 500 bp windows as described in the previous section as a background distribution. We calculated the true positive rate as a function of different RPKM thresholds using TSS or EP300 standards, and calculated the false positive rate as the fraction of background regions above the thresholds. We generated receiver operating characteristic (ROC) curves for PRO-cap detecting TSS and EP300 sites, and compared the PRO-cap ROC with an available cap analysis of gene expression (CAGE) data in GM12878 (RIKEN poly-A-selected nuclear and cytoplasmic CAGE) and available NET-CAGE data in GM12878 (Fig. 1c–e). Subsampling of reads was also performed for a direct comparison between techniques.

**Classification of tTREs into promoters and enhancers.** To classify the tTREs into gene promoters and enhancers/regulatory elements, we used two criteria: (1) distance to annotated refseq TSSs, (2) overlap with CAGE sites. For the refseq based promoter annotations, we defined the tTREs with at least one nTSSs within 500 bp of refseq TSSs as the promoters (n = 14,986). We then defined tTREs greater than 2 kb away from any refseq TSSs as distal enhancers (n = 34,922). For the CAGE based promoter and enhancer classifications, PRO-cap and CAGE counts at the nTSSs are collected for the plus and the minus strands separately, and RPM normalized. nTSSs with CAGE counts significantly above the background were called promoter TSSs. To estimate the background CAGE read counts, we first selected nTSSs that are at least 1 kb away from any annotated refseq TSS (n = 38,658), and calculated the CAGE read counts for both the plus and the minus strands at these background regions. We calculated the p-values of the nTSS CAGE read counts based on the empirical background distribution, and found 13,833 nTSSs that have CAGE readcounts that are significantly higher than the background using FDR < 0.1 (P < 0.0145, CAGE RPM > = 1.18655). Of the 13,833 nTSSs, 995 were bidirectional pairs, yielding 12,878 tTREs as promoters, 995 of which are bidirectional. We defined the remaining 74,948 tTREs as enhancers.

**DNA sequence motif analysis.** We used the RTFBSdb suite[40] that clusters transcription factor binding motifs based on similarity, chooses a representative motif for each cluster based on the expression data, and computes enrichment for known motifs. We filtered CIS-BP motifs based on expression in our LCLs using our PRO-seq data, clustered the motifs with agnes clustering into 400 clusters, and chose a representative motif for each cluster based on expression in our PRO-seq data. We used this motif list to look for motif enrichment within tTRE centers. To assess the cell type specificity of the DNA sequence motifs, we used GTEX[41] RNA-seq data across 54 different cell and tissue types (v1.1.9), which includes LCLs, and calculated the p-value of differential LCL expression compared to the remaining 53 cell types for each gene. We compared the -log10 p-value distributions of all CISBP TFs (background) against TRE enriched CISBP TFs.

**Regulatory variant enrichment analysis.** We used previously defined lists of expression QTLs[22] and DNase I hypersensitivity QTLs[5] in Yoruba LCLs and lifted the coordinates over to the hg19 genome (n = 1,090 and 8,899 respectively). For the disease associated GWAS SNPs, we used the NIH GWAS Catalog[32] entries as of March, 2015 in hg19 (n = 12,239), using all GWAS SNPS as well as SNPs filtered for B-cell related effects (n = 614, Supplementary Table 3). We used the refseq annotation based distal enhancer and promoter classifications for the tTREs, and calculated the per base QTL frequency relative to the tTRE midpoints. For

comparison, we also generated enrichment plots relative to the DNase I hypersensitivity sites (ENCODE DHS; $n = 52,292$ promoter DHS, 226,832 distal enhancer DHS) in all LCL cell lines. (Supplementary Fig. 2c). We also calculated enrichment of these SNPs in tTREs by using the Broad Institute tool SNPsnap[36] to identify matched SNPs for background expectations, then found the overlap of the SNPs of interest and background SNPs with the tTREs. A two-sided Fisher's exact test was used to compute $P$-values for each SNP category.

**Co-expression analysis between tTRE and mRNA**. We used the RNA-seq expression data from Pickrell et al.[22], and selected 13,002 genes with the mean expression level greater than 1 RPKM. 275,660 pairs of tTREs and annotated mRNA TSS within 1 Mb were tested for covariation between tTRE nascent transcription and mRNA expression, and the Pearson correlation coefficients of the 67 individual samples were calculated.

**Variant sensitive alignment of PRO-cap reads**. In summary, we reconstructed the individual phased haplotype genomes, masked out any tTRE regions with ambiguous mappability, then re-aligned the PRO-cap reads to the individual haplotype genomes (Supplementary Fig. 3a, b). Because of removing allele mappability biased regions, we re-calculated read per million (RPM) of the read counts sum across all individuals for each tTRE, and used a RPM cut-off of 0.5 to further select tTRE peaks for testing associations ($n = 76,630$). See supplement for details.

**Normalization of transcription initiation phenotypes**. First, we normalized the read counts to the sequencing depth. We added the plus and the minus strand read counts in each tTRE window, then divided the raw read counts by per million total read counts in the tTRE windows for each individual. Then we used a quantile normalization, where the distribution of read counts in an individual is matched to a reference distribution. For the reference distribution, we applied a median of the ratio normalization method that is used in the DEseq RNA-seq analysis software[42] which finds the ratio of each gene's read count to the mean of all read counts for each gene across all samples and then uses that to scale the individual read counts. We used this median of the ratio normalized distribution for the quantile normalization of the read counts (Supplementary Data 2).

**Identification of variably expressed tTREs**. To identify tTREs that are variably expressed, we used normalized PRO-cap read count data that contain partial replicates ($n = 8$; Supplementary Fig. 3c). For each tTRE, we calculated the deviation from the mean of the normalized read counts between replicates and between different samples. Then we used a one-sided Wilcoxon's rank sum test to test the alternative hypothesis that the differences between samples are greater than between the replicates for each tTRE, and calculated $p$-values. We estimated the number of variably expressed tTRE by analyzing the complete distribution of the $p$-values as described previously[43]. Under the null hypothesis, $p$-values should have a uniform distribution with a density of 1, but the observed $p$-values are only uniformly distributed for large $p$-values. The density of the portion of the $p$-value distribution that is uniform is ~0.281, indicating that up to ~71.9% of tTREs can be considered variably expressed. Using FDR < 0.2, we identified 29,694 variably expressed nTSSs.

**Transcription initiation QTL (tiQTL) association testing**. We tested the association between the nascent transcription initiation phenotypes at each tTRE region and the genotype of SNPs and short indels within a cis range of 2 kb from the midpoint of the nTSS regions. We took the variant sensitive normalized PRO-cap readcounts, and identified tiQTLs using the MatrixEQTL R package[44]. We used allele-mappability-bias-masked tTRE regions on autosomes ($n = 76,118$), and variant sites with a minor allele frequency greater than 5% in our haplotype-phased individuals ($n = 9,808,709$). A total of 994,993 pairs were tested. We tested up to 20 principal components (PCs) as co-variates in 2 kb cis-tiQTL tests and found that 16 PCs gave the largest number of significant tiQTLs (FDR > 0.1). In total 16,193 tTREs are associated with at least one tiQTL within a 2 kb cis region.

**Directional initiation QTL (diQTL) association testing**. We tested the association between the relative direction of the divergent bidirectional nascent transcription initiation pairs at the two nTSSs in each tTRE, and the genotypes of the genetics variants. To compute a metric for the directionality of tTREs, we calculated the directionality index as ratio of plus-strand (1–250) and minus-strand (−250 - 0) read counts, $log_2$ transformed the ratio and quantile normalized the resulting index using the same method we used for diQTLs. The association of these directionality phenotypes with genotype was assessed, as for tiQTLs, using MatrixEQTL. We determined that using 8 principal components as covariates in association testing gave the largest number of significant associations.

**Local-minimum $p$-value filtering for more likely primary SNPs**. We split the genome into 5 kb windows, staggered by 1 kb so that each SNP is in 5 different 5 kb windows. In each window we keep only the SNP with the lowest $P$-value,

independent of which tTRE or gene it affects. If there were two or more SNPs with the lowest $P$-value, none were kept.

**Measuring the effect of tiQTL SNPs on TF sequence motifs**. We tested the effect of tiQTLs on TF binding likelihood as described by Degner et al.[5]. We limited our analysis to the tiQTLs at the center-region (midpoint ±40 bp) of tTREs and imposed a stricter filtering requirement by keeping only those QTLs where the most significantly associated SNP has a $P$-value an order of magnitude lower than the next most significant ($n = 1,213$). Reference and alternative alleles were categorized into enhancing and repressive alleles based on the relative PRO-cap readcounts around the tTREs. We used CIS-BP human transcription factor frequency matrices to generate position weight matrices (PWMs), and queried a 40 bp region surrounding each tiQTL for strong matches to the PWMs (motif score > 13). We repeated this analysis for both alleles and compared the resulting motif scores for enhancing and repressive alleles. We then took these motif score differences in these strong-matched TF motifs to obtain the overall effect of each variation on TF binding potential. For comparison, we also performed this analysis on dsQTLs[5] that fall within the most highly affected window (100 bp windows) ($n = 967$) and dsQTLs outside the affected windows ($n = 7,140$).

**Identifying enrichment among motifs disrupted by tiQTLs**. We used motifbreakR[45] to calculate the motif disruption score for each tiQTL for the 400 motifs selected from the CIS-BP motif database with RTFBSdb. A motif is considered disrupted by the tiQTL if there is a significant match to a motif in at least one allele ($p < 0.01$) and there is at least a 0.5 bit difference in motif matching scores between the alleles. We then count the number of times each motif is disrupted by a tiQTL and compare it to the number of times that same motif is disrupted by randomly selected SNPs within the same region (±200 bp from tTRE midpoint).

**Measuring the effect of diQTL SNPs on Initiator elements**. Initiator (Inr) element likelihood is calculated using published human Inr frequency matrix[37] by taking the natural exponent of the PWM scores. First, we selected diQTL SNPs within 5 base pairs (bp) from the defined nascent transcription start sites (nTSSs) of tTREs ($n = 209$). We calculated the strand specific Inr likelihood difference between the two alleles and included diQTLs that generate differences in Inr likelihood score (>5). We oriented the diQTL directionality effect sizes towards the strand in which the Inr scores are calculated, and assigned the Inr likelihood difference as (high directional allele − low directionality allele). For comparison, we selected tiQTL SNPs within 5 bp from the center of tTREs ($n = 520$) and dsQTLs at the nTSS ($n = 85$) and at center ($n = 71$), calculated Inr likelihood differences in between relatively higher directional allele to the lower directional allele towards the direction of the Inr element, and plotted the distribution of ΔInr likelihood similar to diQTLs.

**Measuring the effect of diQTL SNPs on TATA-like elements**. We focused on the area 40–20 bp upstream of tTRE TSSs and looked for instances of TATA, or inversions thereof, at diQTLs ($n = 162$) as well as tiQTLs at center ($n = 520$) as described above as well as dsQTLs in this area ($n = 108$) and at center ($n = 71$). We oriented the diQTLs as explained for Inr and calculated a delta TATA score (high directionality–low directionality allele). A perfect TATA got a score of 4, a single inversion (A − > T, T − > A) such as TTTA or TAAA got a score of 2, and two inversions got a score of 1. Anything else scored 0.

**Expression QTL association testing**. Using only SNPs that fall within 2 kb of tTREs (same range as the ti- and diQTLs, $n = 19,666,443$), we tested the association between SNPs and gene expression. By limiting our analysis only to the region we are interested in, we increase our power to detect significant associations. We used RNA-seq data from Pickrell et al.[22], merged replicates by taking an average across replicates for each gene, and performed the same quantile normalization as above. We used matrixEQTL to identify eQTLs within a cis-distance of 200 kb (FDR < 0.05). We determined that using 13 PCs as covariates gave the largest number of significant associations.

**Overlap between eQTL- and diQTL/tiQTL-containing tTREs**. We identified enhancers that contain associated diQTLs and/or tiQTLs within 200 bps of the enhancer midpoint. We then computed the proportion of those that overlap an eQTL and compared that ratio with the proportion of enhancers without such QTLs that overlap an eQTL using GenomicRanges.

**Overlap between diQTLs/tiQTLs and other QTLs**. We used previously mentioned lists of expression QTLs ($n = 1,090$), DNase I hypersensitivity QTLs ($n = 8,899$), and disease associated GWAS hits for all traits and B-cell related traits ($n = 12,239$ and $n = 614$ respectively). For the transcription factor binding QTLs, we used published QTLs[26] for JunD, NF-κB, Pou2f1, PU.1, and Stat1 ($n = 157,235$, 109,797, 102,105, 94,398, and 12,224 respectively). We overlapped these QTL lists with filtered diQTL and tiQTL lists ($n = 4,479$ and $n = 22,949$ respectively), as well as background lists ($n = 383,347$). Background SNPs were all SNPs within 2 kb of TRE centers. We then computed the percentage of diQTL, tiQTL, or background

SNPs that overlapped and found the enrichment of diQTL and tiQTL overlap over background. We used two-sided Fisher's exact tests on the overlap-count data to compute $p$ values. diQTLs at the TSS ($n = 206$) were ±20 bp from the TSS of the TRE, and tiQTLs at the TRE center ($n = 1230$) were ±25 bp from the TRE center. Background SNPs for center and TSS were filtered similarly ($n = 11,023$ and $n = 8716$ respectively).

**Gene expression association of tiQTLs/diQTLs**. We computed a $p$-value for the association of each enhancer tiQTL and diQTL with gene expression using matrixEQTL and a cis-distance of 200 kb, reporting all $p$-values. We then compared the distribution of $p$-values ($-\log10$) with those for all SNPs within 2 kb of tTREs using QQ plots. We also split diQTLs into groups of diQTLs which are not tiQTLs ($n = 4326$), diQTLs±25 bp from the TRE center ($n = 284$), and diQTLs±25 bp from the nTSS of TREs ($n = 326$) and compared those to all SNPs. We then found the percentage of SNPs which pass an FDR threshold of 0.1 with a Benjamini–Hochberg correction and compared each group to all SNPs.

**Regional eQTL enrichment analysis**. Enhancers were split into regions according to Fig. 5c. Center is ±25 bp from the midpoint and core is ±25 bp from the TSS. The number of eQTLs in each region was counted and normalized to the total number of SNPs in the region.

**Reporting summary**. Further information on research design is available in the Nature Research Reporting Summary linked to this article.

## Data availability
The sequencing libraries from this study have been submitted to the NCBI Gene Expression Omnibus (GEO) with accession number GSE110638. Yoruba DNase-seq data and dsQTLs were previously deposited by Degner et al[5] at GEO with accession number GSE31388. RNA-seq data were deposited by by Pickrell et al.[22] in GEO under accession number GSE19480. Data from the Genotype-Tissue Expression (GTEx) project[41] which was used for analysis in this manuscript was obtained from the GTEx Portal. NET-CAGE data used for analysis was deposited by Hirabayashi et al[28] at GEO under accession number GSE118075. ENCODE RIKEN CAGE data was obtained from the UCSC Genome Browser. All transcription factor binding QTLs came from supplementary information in Tehranchi et al.[26] hg19 reference sequence and RefSeq annotations were obtained from the human genome resources at NCBI. GWAS SNPs used for analysis were obtained from the NIH GWAS Catalog[32] in March 2015. EP300 sites were obtained from ENCODE Factorbook with ENCODE accession number ENCSR000DZD. The source data underlying Figs. 1g and 5d and Supplementary Figs. 2i and 7ab-d, f, h are provided as a Source Data file. The data supporting the findings of this study are available from the corresponding authors upon reasonable request.

## Code availability
All custom code used for analysis which is central to the conclusions of the paper is available on GitHub at https://github.com/hyunminkang/eRNA_YRI_paper_code. All other code is available upon request.

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

## Acknowledgements

We thank Dr. Jun Hee Lee from the University of Michigan, Dr. John Lis and Dr. Charles Danko from Cornell University, and members of their labs, for helpful discussions during the inception and completion of this work. This work was supported by a Cornell University Center for Vertebrate Genomics (CU-CVG) scholars award (to K.K.) and NIH U01HL137182 (to H.M.K.).

## Author contributions

Project was conceived by H.K. and H.M.K. Cell culture, library preparations, sequencing, and processing of raw data were done by H.K. in H.M.K.'s lab. K.K., A.D. performed the analyses and generated figures. K.K., A.D., H.K., H.M.K. wrote the manuscript.

## Competing interests

The authors declare no competing interests.
