## [Peer Review File · Nature Communications]

Reviewers' comments:

Reviewer #1 (Remarks to the Author):

This manuscript describes that PRO-cap permits a high-resolution analysis of the activity and architecture of transcribed transcriptional regulatory elements (tTRE) with improved detection sensitivity compared to other transcriptome analyses. The analysis demonstrated a significant association of genetic variants with enhancer and promoter transcription activity. The association was more prominent at enhancers than promoters. Intriguingly, the patterns of genetic variant enrichment largely fall into two distinct categories, transcription factor binding sites (TFBS) located in the center of regulatory regions and the transcription core element-like regions in the vicinity of transcription start sites that affect transcription directionality. Both variant types appear to be related to gene regulation as both tiQTLs and diQTLs are enriched for eQTLs. Taken together, this molecular dissection study provides compelling evidence that the bipartite architecture of enhancers is an important feature of enhancer function and also raises the possibility that eRNA transcription activity might be functionally linked to target gene regulation. The major findings described in the manuscript are valuable information that can benefit the broad field in molecular biology and thus recommended for publication in Nat Comm. Below are minor comments for the authors to consider/clarify.

1. Are the changes (increase or decrease) in tTRE activity positively correlated with expression changes in nearby genes?
2. The analyses described in fig. 5 and suppl fig. 6 show that transcription-associated QTLs correlate with gene expression. This was performed by analyzing correlative changes between tTRE activities and genes within the 200kb window regardless of their relative locations. In the case of diQTL association with gene expression change, it might be useful to know whether the relative position of those genes with altered transcription activity can be correlated with the altered directionality in diQTL.

Reviewer #2 (Remarks to the Author):

In this work, Kristjánsson et al. profile capped, nascent RNA using PRO-cap (n=70 samples) and PRO-seq (n=10 samples) in Yoruba lymphoblastoid cell lines (LCLs). The authors identify promoter and enhancer transcribed transcriptional regulatory elements (tTREs), observe their overlap with other regulatory annotations and relevant genetic variants such as eQTL and GWAS loci. The authors then identify QTL for transcription initiation and directionality and investigate the patterns of QTL overlap in tTREs and underlying motifs. While PRO-cap and PRO-seq profiles present a highly relevant and useful dataset of transcription initiation, especially at enhancers in LCLs, several analyses in the paper call for a more robust methodology and/or require more clarification:

1. Reads from across samples were merged while identifying tTREs. Was sequencing depth accounted for?

2. In Fig 1C shows ROC curves for selected samples for each technique PRO-cap (GM19223), CAGE (GM12814 nuc), NET-CAGE(GSM3318219). Given the 69 samples for which PRO-cap was performed, it would be more informative if ROC curves were also generated for aggregate PRO-cap profiles. Aggregate profiles for other techniques for which multiple samples are available would also be useful to compare. The selected samples are not matched across the techniques, if data across multiple techniques are available on the same cell line, comparing those would be more appropriate. Fig 1D shows the area under the ROC curves for n=6 PRO-Cap or n=4 CAGE data - it is unclear how the respective datasets were selected and here, and why not all PRO-cap samples that were profiled included.
3. Fig 2C shows the overlap fraction of tTREs with regulatory regions and vice versa. While this information is useful, given the difference in sizes of the regulatory elements considered, it would be more informative to compute overlap enrichment - i.e. observed overlap relative to a background expectations. Tools such as genomic annotation test (GAT) DOI 10.1093/bioinformatics/btt343 might be useful here.
4. Line 108-110 the authors mention that “tTREs contain ENCODE annotated TFBSs, and the most enriched TFBSs tend to be cell-type relevant”. The referenced supplementary fig 2B shows the top five “enriched” motifs. This enrichment analysis calls to be better presented for a more quantitative interpretation, eg. showing enrichment scores or p values across the motifs tested in promoter vs enhancer tTREs, possibly clustering and highlighting the set of cell-type relevant motifs.
5. Fig 2D: eQTL and dsQTLs datasets used in this analysis corresponded to what tissue types? Were eQTL/dsQTLs in LCL-unrelated control cell types also considered for comparison? Lines 110-113 and 396-398 are missing appropriate citations. Also, GWAS data for what traits were chosen? Were GWAS for traits relevant for LCL cell-type and some unrelated control traits included? Such information is imperative to appropriately infer these analyses.
6. Lines 110-113 mention “the identified enhancers are enriched with regulatory information”, however, the analysis in Fig 2D shows the eQTL/GWAS SNP frequency per bp and not technically ‘enrichment’. To test if regulatory variants overlap tTRE’s more often than just expected by chance, one should also consider a set of control SNPs, matched for the minor allele frequency, number of proxy SNPs, distance to the nearest gene etc. parameters and then compare overlap with the genomic annotations. Also, since the eQTL/GWAS association analyses tag multiple variants in high linkage disequilibrium (LD), were proxy SNPs also considered for these analyses? A number of publicly available tools can be useful for computing enrichment of regulatory variants in genomic annotations: DOI 10.1038/s41588-018-0322-6, 10.1016/j.ajhg.2014.03.004, 10.1038/ng.3404 etc.
7. In the QTL analyses, were the library preparation batches, sample collection, sex etc. accounted for?
8. The authors report that 16,193 tTREs have an associated tiQTL (FDR < 0.1). How many unique tiQTL loci does that correspond to? Some QC of the QTL association analysis such as qq plots would be useful here. A previous study using CAGE in 154 lymphoblastoid cell lines identified 5376 promoter usage and 110 enhancer activity QTLs. It could be quite useful to compare QTL loci between the two sets, given the similar cell type and transcription initiation profiling, albeit using different methods. It might be that CAGE identifies a subset of more stably transcribed enhancers whereas PRO-cap and PRO-seq identify a larger set of enhancers.
9. In Figs 3B,C, it is unclear to me what is meant by ‘in target’ and ‘outside target’. Also, it is unclear how are the authors accounting for LD proxy SNPs of the tiQTL lead SNPs while computing SNP frequencies

relative to tTRE midpoints. Depending on the QTL loci, lead SNPs might not be 'causal' and proxies should be considered.

10. In Figs 3B,C the authors show tiQTL density relative to TRE center and compare to background density from all SNPs in the region. Since QTL likely overlap accessible chromatin regions where transcription factors bind, it would be informative to overlay aggregate DNase or ATAC-seq tracks to see if chromatin accessibility profiles show a similar pattern at enhancer vs promoter regions.

Other comments:

1. Fig 1B: What are the normalization factor and y range for each of the tracks shown?
2. Fig 1C: For CAGE, does 'ENCODE cell' mean CAGE library prep from cellular RNA and 'GM12814 nuc' mean CAGE library prep from nuclear RNA? This could be clarified in the figure legend/caption. Also, which cell line is the ENCODE data for?
3. Lines 143-145 are unclear - what are the numbers of dsQTL?
4. Figure 3B, C show x axis labels as "relative position from enhancer midpoint" or "relative position from promoter midpoint", however from the text (line 173) it seems that tiQTL density around TRE midpoints was assessed. 'Promoter TRE' or 'Enhancer TRE' would be more appropriate labels.
5. The authors use the directionality index as $\log_2(\text{plus reads}/\text{minus reads})$ and identify more directionality QTL at promoters than enhancers. I wonder if because promoters are more directional, D could attain quite extreme values. The FANTOM work defined D as $(\text{plus reads} - \text{minus reads})/(\text{plus reads} + \text{minus reads})$ which results in D values between -1 and 1.

Reviewer #3 (Remarks to the Author):

Here Kristjansdottir et al present a study on how genetic variants affect eRNA transcription in lymphoblastoid cell lines from 67 individuals. Through their analysis they identify variants that impact the level of transcription and the directionality. The analysis also provides insights regarding the structure of enhancers. Overall, I think that this is a very interesting study which adds a new and important perspective, variation across individuals, to the study of eRNAs. The paper is also well-written and it is easy to follow the arguments presented. Although the paper is well suited for Nature Communications, I believe that the following major issues need to be addressed:

* In fig 3e the authors claim that there are higher PWM scores for alleles with high levels of transcription. However, they need to report a suitable statistical test and p-value to support this claim. The same holds for fig 4e, f.

* Fig 4d is very nice and convincing, but it needs more information as there are no scales on the y-axis. I would like to have quantitative information regarding the levels of transcription for different genotypes. In addition, I would also like to see reports on the significance of the reported differences.

* I would like to get a better sense about the similarities (and conversely the differences) between

diQTLs and tiQTLs. In particular, it is not clear how often they are found at the same enhancer or promoter? Is co-occurrence enriched or depleted compared to a null model? Are there some variants that come up as both diQTLs and tiQTLs? When found together, are they distributed in a different manner than in fig 4b,c.

* On line 251 the authors report an enriched overlap of di/tiQTLs with eQTLs. However, I believe that this enrichment is somewhat misleading as the predictive power that is provided by finding a di/tiQTL is very low. If I understand fig s6a correctly, the probability of finding an eQTL goes from ~1% to 2-4%. This is something that needs to be commented on in the main text. A similar point holds for the discussion in the last paragraph before the discussion where it should be made clear that the absolute levels are very low.

* I am not entirely impressed by the violin plots of p-values in figs S6b-d. Even though the distributions are almost identical, with extremely small differences in medians, the authors report p-values south of 10^{-15} . This suggests to me that an inappropriate test has been applied as the magnitude of effect in terms of the eQTL p-values between the different categories are virtually indistinguishable. The authors need to present a valid analysis which considers not just p-values but also magnitude of effect to back up the claims in the text

I also would like to see the authors fix the following minor issues:

* In fig 1a it would be helpful if the AUROC values were reported in the legend

*One of the main challenges in studying eRNAs have been to establish their function. I am happy to see that the authors discuss this and I would like them to add some speculation on what kinds of experiments could be conducted in the future given their findings to get at this question.

Reviewer comments:

Reviewer #1 (Remarks to the Author):

This manuscript describes that PRO-cap permits a high-resolution analysis of the activity and architecture of transcribed transcriptional regulatory elements (tTRE) with improved detection sensitivity compared to other transcriptome analyses. The analysis demonstrated a significant association of genetic variants with enhancer and promoter transcription activity. The association was more prominent at enhancers than promoters. Intriguingly, the patterns of genetic variant enrichment largely fall into two distinct categories, transcription factor binding sites (TFBS) located in the center of regulatory regions and the transcription core element-like regions in the vicinity of transcription start sites that affect transcription directionality. Both variant types appear to be related to gene regulation as both tiQTLs and diQTLs are enriched for eQTLs. Taken together, this molecular dissection study provides compelling evidence that the bipartite architecture of enhancers is an important feature of enhancer function and also raises the possibility that eRNA transcription activity might be functionally linked to target gene regulation. The major findings described in the manuscript are valuable information that can benefit the broad field in molecular biology and thus recommended for publication in Nat Comm. Below are minor comments for the authors to consider/clarify.

We thank reviewer 1 for the excellent comments and suggestions, in particular regarding utilizing the co-expression network approach to explore connections between enhancer tTREs and gene expression. Since the co-expression network approach is independent of and parallel to the genetic analysis that we focused on the current manuscript, we are in the process of further developing this idea into an independent study.

Q1-1) Are the changes (increase or decrease) in tTRE activity positively correlated with expression changes in nearby genes?

A1-1) As anticipated, we do see that tTRE activity is positively correlated with the expression in nearby genes. When we plotted the correlation coefficients of the PRO-cap read counts between two tTREs (including enhancers and promoters) across our LCL samples, we observed a stronger positive correlation when the tTREs are nearby (up to 100 kb; Figure c). We also observed positive correlation between PRO-seq read counts of tTRES and mRNA expression levels using RNA-seq data from Pickrell et al (Figure d). This correlation decays over increasing distance. While the expression variation is within the same cell type (LCLs) and smaller than comparing between different cell types, our PRO-cap analysis shows the sensitivity to capture these distance dependent positive correlations. We have added the tTRE-mRNA covariation analysis to our manuscript in Supplementary Figure 3D.

Q1-2) The analyses described in fig. 5 and suppl fig. 6 show that transcription-associated QTLs correlate with gene expression. This was performed by analyzing correlative changes between tTRE activities and genes within the 200kb window regardless of their relative locations. In the case of diQTL association with gene expression change, it might be useful to know whether the relative position of those genes with altered transcription activity can be correlated with the altered directionality in diQTL.

A1-2) This is an insightful suggestion that will allow us to better understand the mechanistic basis of eRNA directionality and its function. We were not able to detect a statistically significant correlation between diQTL directionality and position of the nearby associated gene/promoter. This is likely due to the relatively lower number of diQTLs, which limits the power of the analysis. However, when we disregard the genotype information and utilize a co-expression network approach instead, we observe a stronger positive correlation between PRO-cap read counts of convergent and sense strands than divergent strands when examining pairs of nearby tTREs (see figures). This leads to the anticipation that diQTLs that shift the directionality of the eRNA towards the gene promoter will be positively associated with gene expression level.

Reviewer #2 (Remarks to the Author):

In this work, Kristjánsson et al. profile capped, nascent RNA using PRO-cap (n=70 samples) and PRO-seq (n=10 samples) in Yoruba lymphoblastoid cell lines (LCLs). The authors identify promoter and enhancer transcribed transcriptional regulatory elements (tTREs), observe their overlap with other regulatory annotations and relevant genetic variants such as eQTL and GWAS loci. The authors then identify QTL for transcription initiation and directionality and investigate the patterns of QTL overlap in tTREs and underlying motifs. While PRO-cap and PRO-seq profiles present a highly relevant and useful dataset of transcription initiation,

especially at enhancers in LCLs, several analyses in the paper call for a more robust methodology and/or require more clarification:

We greatly appreciate the second reviewer's comments and suggestion to add more robust analysis and clarification of the methods. We performed more detailed analysis in response to the reviewer's comments and attempted to provide as much information as possible, which we believe has greatly improved our revised manuscript.

Q2-1-1) Reads from across samples were merged while identifying tTREs. Was sequencing depth accounted for?

A2-1-1) The sequencing depths for individual samples were not directly accounted for in our initial analysis. Our strategy was to first use all merged reads to maximize the sensitivity of tTRE discovery, under the assumption that tTREs of the LCLs are the same between individuals and the individual differences are in the level of tTRE expression. We then selected the tTREs more stringently using additional criteria such as allele mappability biases as described and expression level cut-offs. An increased read count increases the number of tTREs detected due to the emergence of bidirectional PRO-cap peak patterns (see figure). This indicates that lowly transcribed enhancer or promoter elements may still exist that are under the detection threshold and supports our use of merged reads to maximize discovery.

Q2-1-2) In Fig 1C shows ROC curves for selected samples for each technique PRO-cap (GM19223), CAGE (GM12814 nuc), NET-CAGE(GSM3318219). Given the 69 samples for which PRO-cap was performed, it would be more informative if ROC curves were also generated for aggregate PRO-cap profiles. Aggregate profiles for other techniques for which multiple samples are available would also be useful to compare. The selected samples are not matched across the techniques, if data across multiple techniques are available on the same cell line, comparing those would be more appropriate. Fig 1D shows the area under the ROC curves for n=6 PRO-Cap or n=4 CAGE data - it is unclear how the respective datasets were selected and here, and why not all PRO-cap samples that were profiled included.

A2-1-2) We agree that comparison of the techniques in exactly the same cell line would be the most robust and ideal analysis. However, our list of PRO-cap cell lines did not overlap with the available data for the other techniques (CAGE and NET-CAGE). But, all these cell lines are among the lymphoblastoid cell lines (LCLs) produced by the HAPMAP consortium using the same method, and aggregate profiles for all techniques would

be useful. Initially, we displayed the individual cell lines instead of the aggregate profiles, since using all the PRO-cap reads from the aggregate data will bias in favor of PRO-cap due to the higher sequencing depth (1.4 billion unique reads). As suggested, we have changed main figure Fig 1c to include aggregate profiles for the techniques we include. To address read-depth bias, we have also added a comparison between PRO-cap and other techniques where datasets for each of the techniques are sampled to have matched read counts. The new ROC curves were not markedly different from each other nor from the original curves based on individual cell lines.

Q2-1-3) Fig 2C shows the overlap fraction of tTREs with regulatory regions and vice versa. While this information is useful, given the difference in sizes of the regulatory elements considered, it would be more informative to compute overlap enrichment - i.e. observed overlap relative to a background expectations. Tools such as genomic annotation test (GAT) DOI 10.1093/bioinformatics/btt343 might be useful here.

A2-1-3) Thank you for this suggestion. We agree that computing overlap enrichment for these regulatory elements is a useful addition and have added a figure with computed enrichments using the suggested tool. All of the regulatory elements show significant enrichment in tTREs over background expectation as shown below.

Q2-1-4) Line 108-110 the authors mention that “tTREs contain ENCODE annotated TFBSs, and the most enriched TFBSs tend to be cell-type relevant”. The referenced supplementary fig 2B shows the top five “enriched” motifs. This enrichment analysis calls to be better presented for a more quantitative interpretation, eg. showing enrichment scores or p values across the motifs tested in promoter vs enhancer tTREs, possibly clustering and highlighting the set of cell-type relevant motifs.

A2-1-4) As the reviewer suggested, we provide quantitative enrichment scores and p-values for the tTRE enriched TFBS motifs by the RTFBS package in the supplementary materials. We also used GTEX RNA-seq data to quantify LCL specific expression of the transcription factors used in the TFBS motif search, and find

that TFBSs in enhancer and promoter tTREs show significant enrichment of LCL specific expression over all TFs used in the analysis ($p = 0.008, 0.002$ respectively by Wilcoxon test).

Q2-1-5) Fig 2D: eQTL and dsQTLs datasets used in this analysis corresponded to what tissue types? Were eQTL/dsQTLs in LCL-unrelated control cell types also considered for comparison? Lines 110-113 and 396-398 are missing appropriate citations. Also, GWAS data for what traits were chosen? Were GWAS for traits relevant for LCL cell-type and some unrelated control traits included? Such information is imperative to appropriately infer these analyses.

A2-1-5) We apologize for the lack of proper citations and missing information. The eQTL and dsQTL datasets are from the same LCL cell lines in the Yoruba population. No eQTLs and dsQTLs in LCL-unrelated control cells were used as we use different background SNP sets as controls. On the other hand, GWAS data include both cell-type relevant and noncell-type relevant traits. While it would be ideal to use the GWAS for LCL specific traits and compare that to unrelated control traits, the number of known LCL specific traits publicly available were limited (21 B-cell associated diseases or traits, 614 SNPs from the GWAS catalog of 12,239 SNPs overlapped with our analysis). We did observe a 1.88-fold enrichment of the B-cell associated GWAS SNPs in our tiQTL and diQTL lists, however, these did not meet the statistical significance threshold. The text has been changed to include this information and the relevant citations.

Q2-1-6) Lines 110-113 mention “the identified enhancers are enriched with regulatory information”, however, the analysis in Fig 2D shows the eQTL/GWAS SNP frequency per bp and not technically ‘enrichment’. To test if regulatory variants overlap tTRE’s more often than just expected by chance, one should also consider a set of control SNPs, matched for the minor allele frequency, number of proxy SNPs, distance to the nearest gene etc. parameters and then compare overlap with the genomic annotations. Also, since the eQTL/GWAS association analyses tag multiple variants in high linkage disequilibrium (LD), were proxy SNPs also considered for these analyses? A number of publicly available tools can be useful for computing enrichment of regulatory variants in genomic annotations: DOI 10.1038/s41588-018-0322-6, 10.1016/j.ajhg.2014.03.004, 10.1038/ng.3404 etc.

A2-1-6) Thank you for this excellent suggestion. We have added an analysis showing the enrichment of the QTLs and GWAS SNPs over a background matched set of control SNPs in the tTREs. This analysis shows a significant enrichment of all SNP categories in distal tTREs and of dsQTLs and eQTLs in promoter tTREs.

Q2-1-7) In the QTL analyses, were the library preparation batches, sample collection, sex etc. accounted for?

A2-1-7) In our QTL analysis we used PCA analysis to account for possible confounding effects such as batches and sex of the individual, which is a method used in other QTL studies (ex Degner et al. 2012). When covariates for batch and sex were explicitly included for each sample we find that 71.76% of the original unfiltered tiQTLs are found with FDR < 0.1. Additionally, 76.74% of the tiQTLs found using added covariates are found in our original list of tiQTLs. For diQTLs we see a similar percentage, with 73.02% of diQTLs rediscovered and 86.99% of new diQTLs in the original. Furthermore, when the FDR threshold for the tiQTLs and diQTLs found with added covariates is increased to 0.15 we find that the percentage of original QTLs rediscovered increases by about 10% for both diQTLs and tiQTLs and are likely excluded from under the 0.10 FDR cutoff due to small variations between the two analyses. With this high percentage of rediscoveries, we believe that using principal components as covariates works well. Furthermore, looking at our principal components we do not see grouping by batch in the first two PCs (see figures below), indicating that our batch effect is not dominating. As an example of how using PCs as covariates is valuable, we do see a pattern based on the number of unique reads per sample in the first two PCs (see figures below), but that is gone after the first two principal components. The inclusion of those PCs as covariates, therefore, accounts for differences in the number of uniquely mapped reads as a variable in our QTL analyses.

Q2-1-8) The authors report that 16,193 tTREs have an associated tiQTL (FDR < 0.1). How many unique tiQTL loci does that correspond to? Some QC of the QTL association analysis such as qq plots would be useful here.

A2-1-8) Thank you for this excellent suggestion. We have edited the manuscript to indicate the number of tiQTLs and diQTLs before and after filtering. Also, we have added QQ plots for our diQTL and tiQTL analyses.

Q2-1-9) A previous study using CAGE in 154 lymphoblastoid cell lines identified 5376 promoter usage and 110 enhancer activity QTLs. It could be quite useful to compare QTL loci between the two sets, given the similar cell type and transcription initiation profiling, albeit using different methods. It might be that CAGE identifies a subset of more stably transcribed enhancers whereas PRO-cap and PRO-seq identify a larger set of enhancers.

A2-1-9) We agree that a comparison between our QTLs and promoter usage and enhancer activity QTLs could be interesting. However, the CAGE data comes from LCLs of central European descent, while our data comes from the Yoruba population. As the full list of puQTLs and eaQTLs are not publically available, we compared the puQTLs listed as linked to GWAS SNPs in the CAGE study and saw that only 43% of those QTLs are included in the list of SNPs for our population. Due to this difference in the population background, we expect that it would, unfortunately, be less informative to compare these SNPs directly.

Q2-1-10) In Figs 3B,C, it is unclear to me what is meant by 'in target' and 'outside target'. Also, it is unclear how are the authors accounting for LD proxy SNPs of the tiQTL lead SNPs while computing SNP frequencies relative to tTRE midpoints. Depending on the QTL loci, lead SNPs might not be 'causal' and proxies should be considered.

A2-1-10) We agree that "in vs. outside target" is confusing terminology and have changed the figure accordingly. Also, we appreciate the reviewer's concerns regarding LD proxy SNPs and lead vs. causal SNPs. As we are interested in very finely mapping the location of these QTLs, we decided to take a conservative approach and only include QTLs where LD proxy SNPs are very unlikely to confound our analysis; we keep only lead SNPs with an order of magnitude smaller p-value than any SNP within 5 kb. About a third of our tiQTLs clear this threshold. We agree that, though we believe this approach greatly enriches for causal SNPs, not all of them will necessarily be causal and are therefore careful not to make that claim. However, we see no reason why our approach would bias the pattern of enrichment surrounding tTREs. A more complex analysis that would allow us to use more of our tiQTLs in this analysis might increase our signal, or it might mask our signal with non-causal proxy SNPs.

Q2-1-11) In Figs 3B,C the authors show tiQTL density relative to TRE center and compare to background density from all SNPs in the region. Since QTL likely overlap accessible chromatin regions where transcription factors bind, it would be informative to overlay aggregate DNase or ATAC-seq tracks to see if chromatin accessibility profiles show a similar pattern at enhancer vs promoter regions.

A2-1-11) Thank you for this useful suggestion. DNase-seq tracks have been added to Figs 3B,C and the equivalent panels in Fig 4. DNase-seq tracks for both promoter TREs and enhancer TREs show peaks in the middle of the TRE. At enhancer TREs this closely matches the peak we see for tiQTLs, while at promoter TREs we see that the tiQTL peak is shifted slightly compared to the DNase-seq peak.

Other comments:

Q2-2-1) Fig 1B: What are the normalization factor and y range for each of the tracks shown?

A2-2-1) We have added the y range and normalization factor into the figure. Thank you for pointing out this information was missing.

Q2-2-2) Fig 1C: For CAGE, does 'ENCODE cell' mean CAGE library prep from cellular RNA and 'GM12814 nuc' mean CAGE library prep from nuclear RNA? This could be clarified in the figure legend/caption. Also, which cell line is the ENCODE data for?

A2-2-2) Thank you for pointing out the unclear information in this figure. We have changed the figure legend to indicate the cell line and RNA preparation method used.

Q2-2-3) Lines 143-145 are unclear - what are the numbers of dsQTL?

A2-2-3) Thank you for pointing out this need for clarification. We have edited the text to indicate that we meant the proportion of QTLs at promoters was similar in each set of QTLs.

Q2-2-4) Figure 3B, C show x axis labels as “relative position from enhancer midpoint” or “relative position from promoter midpoint”, however from the text (line 173) it seems that tiQTL density around TRE midpoints was assessed. ‘Promoter TRE’ or ‘Enhancer TRE’ would be more appropriate labels.

A2-2-4) We agree that this wording is potentially misleading and have changed the figure accordingly.

Q2-2-5) The authors use the directionality index as $\log_2(\text{plus reads}/\text{minus reads})$ and identify more directionality QTL at promoters than enhancers. I wonder if because promoters are more directional, D could attain quite extreme values. The FANTOM work defined D as $(\text{plus reads} - \text{minus reads})/(\text{plus reads} + \text{minus reads})$ which results in D values between -1 and 1.

A2-2-5) We compared these two different directionality indexes and found that they have a Pearson correlation of 0.94. When we used a quantile-normalized D from the FANTOM work to identify diQTLs with an FDR > 0.1, 83.4% of our original unfiltered diQTLs are rediscovered. Conversely, 77.15% of the FANTOM D QTLs are identified using our original directionality index. The two groups also have a similar ratio of QTLs found at promoters vs enhancers: 70.82% vs. 69.07% of diQTL containing TREs being enhancers using ours vs. FANTOM versions of D, respectively. The similarity of these percentages suggests that the potentially extreme values of our directionality index are not the cause for the observed enrichment of diQTLs at promoters.

Reviewer #3 (Remarks to the Author):

Here Kristjansdottir et al present a study on how genetic variants affect eRNA transcription in lymphoblastoid cell lines from 67 individuals. Through their analysis they identify variants that impact the level of transcription and the directionality. The analysis also provides insights regarding the structure of enhancers. Overall, I think that this is a very interesting study which adds a new and important perspective, variation across individuals, to the study of eRNAs. The paper is also well-written and it is easy to follow the arguments presented. Although the paper is well suited for Nature Communications, I believe that the following major issues need to be addressed:

We thank the third reviewer for the suggestions to add clarifications of analyses and additional statistical information to the manuscript. We have added additional statistics and clarifications of our claims as well as a new analysis, which we think has strengthened our manuscript.

Q3-1-1) In fig 3e the authors claim that there are higher PWM scores for alleles with high levels of transcription. However, they need to report a suitable statistical test and p-value to support this claim. The same holds for fig 4e, f.

A3-1-1) Thank you for pointing out the necessity of a statistical test to support this claim. We have added p-values (unpaired two-sided Wilcoxon test) evaluating the hypothesis that the delta PWM is larger than zero. In all three panels the test is consistent with our claims. In Fig 3E we include this test as well as a paired test between the tiQTLs and each of the dsQTL comparison groups.

Q3-1-2) Fig 4d is very nice and convincing, but it needs more information as there are no scales on the y-axis. I would like to have quantitative information regarding the levels of transcription for different genotypes. In addition, I would also like to see reports on the significance of the reported differences.

A3-1-2) We agree that adding more information is important in this figure. The scale for transcription has been added to the figure, as well as the mean directionality index for each of the different genotypes. We see a significant difference between the directionality index for each of the three genotypes at that SNP.

Q3-1-3) I would like to get a better sense about the similarities (and conversely the differences) between diQTLs and tiQTLs. In particular, it is not clear how often they are found at the same enhancer or promoter? Is co-occurrence enriched or depleted compared to a null model? Are there some variants that come up as both diQTLs and tiQTLs? When found together, are they distributed in a different manner than in fig 4b,c.

A3-1-3) Thank you for suggesting this important addition to the paper. We have expanded our discussion of shared ti/diQTL variants. We see that the co-occurrence of filtered diQTLs and tiQTLs at the same tTRE is enriched compared to a matched background of SNPs within a tTRE, with a fold enrichment of 3.76 for diQTL tTREs with tiQTLs and 2.48 for the inverse (two-sided Fisher's exact $P < 2.2e-16$ for both). Many of these SNPs are also shared variants: 32.84% of filtered diQTLs are also tiQTLs, and 6.41% of filtered tiQTLs are also diQTLs. When we look at the distribution of these shared SNPs we see a pattern similar to diQTLs, though the pattern is less clear due to the reduced number of QTLs.

Q3-1-4) On line 251 the authors report an enriched overlap of di/tiQTLs with eQTLs. However, I believe that this enrichment is somewhat misleading as the predictive power that is provided by finding a di/tiQTL is very low. If I understand fig s6a correctly, the probability of finding an eQTL goes from ~1% to 2-4%. This is something that needs to be commented on in the main text. A similar point holds for the discussion in the last paragraph before the discussion where it should be made clear that the absolute levels are very low.

A3-1-4) We thank the reviewer for pointing out the confusing representation of this data. We did not intend to claim that the presence of ti/diQTLs strongly predicts the presence of an eQTL. Rather, we point out this enrichment as indirect support for our claim that ti/diQTLs may affect gene expression. We don't necessarily expect a large proportion of di or tiQTLs to have a strong enough effect on gene expression to be detected in an eQTL analysis. We have adjusted the text accordingly.

Q3-1-5) I am not entirely impressed by the violin plots of p-values in figs S6b-d. Even though the distributions are almost identical, with extremely small differences in medians, the authors report p-values south of 10^{-15} . This suggests to me that an inappropriate test has been applied as the magnitude of effect in terms of the eQTL p-values between the different categories are virtually indistinguishable. The authors need to present a valid analysis which considers not just p-values but also magnitude of effect to back up the claims in the text

A3-1-5) Thank you for this excellent comment. We agree that the visualization of these p-values is misleading and have replaced them with QQ plots, along with Kolmogorov-Smirnov test p-values which indicate that the CDF of negative log₁₀ p-values of all SNPs is to the left of the CDFs of di- and tiQTLs. We have also added an analysis of the proportion of SNPs in each category that are identified as eQTLs below an FDR threshold of 0.1; Fisher's exact tests comparing di- and tiQTLs to all SNPs indicate that diQTLs and tiQTLs are more likely to be significantly associated with gene expression.

I also would like to see the authors fix the following minor issues:

Q3-2-1) In fig 1a it would be helpful if the AUROC values were reported in the legend.

A3-2-1) Thank you for this suggestion. We have added the AUROC values to the legend of the figure.

Q3-2-2) One of the main challenges in studying eRNAs have been to establish their function. I am happy to see that the authors discuss this and I would like them to add some speculation on what kinds of experiments could be conducted in the future given their findings to get at this question.

A3-2-2)

This is an excellent suggestion. We have expanded our discussion of further experiments in our discussions section. Briefly, we suggest using reporter assays and/or genome editing to dissect enhancers that show directional changes in our analysis and explore the resulting effects on associated genes.

REVIEWERS' COMMENTS:

Reviewer #1 (Remarks to the Author):

The authors adequately addressed the comments raised by this reviewer. It is recommended for publication.

Reviewer #2 (Remarks to the Author):

The authors have largely addressed my comments and I appreciate the added clarifications and analyses in what I believe is a more robust manuscript.

Reviewer #3 (Remarks to the Author):

The authors have addressed all of my comments and I recommend that the article is accepted for publication.